# Considering Image Information and Self-Similarity: A Compositional Denoising Network

**DOI:** 10.3390/s23135915

**Published:** 2023-06-26

**Authors:** Jiahong Zhang, Yonggui Zhu, Wenshu Yu, Jingning Ma

**Affiliations:** 1The State Key Laboratory of Media Convergence and Communication, Communication University of China, Beijing 100024, China; jiahongz.2023@gmail.com; 2The School of Data Science and Media Intelligence, Communication University of China, Beijing 100024, China; mjn1016230683@163.com; 3School of Optoelectronic Science and Engineering, University of Electronic Science and Technology of China (UESTC), Chengdu 610054, China

**Keywords:** image denoising, residual learning, CNN

## Abstract

Recently, convolutional neural networks (CNNs) have been widely used in image denoising, and their performance has been enhanced through residual learning. However, previous research mostly focused on optimizing the network architecture of CNNs, ignoring the limitations of the commonly used residual learning. This paper identifies two of its limitations, which are the neglect of image information and the lack of effective consideration of image self-similarity. To solve these limitations, this paper proposes a compositional denoising network (CDN), which contains two sub-paths, the image information path (IIP) and the noise estimation path (NEP), respectively. IIP is trained via an image-to-image method to extract image information. For NEP, it utilizes image self-similarity from the perspective of training. This similarity-based training method constrains NEP to output similar estimated noise distributions for different image patches with a specific kind of noise. Finally, image information and noise distribution information are comprehensively considered for image denoising. Experimental results indicate that CDN outperforms other CNN-based methods in both synthetic and real-world image denoising, achieving state-of-the-art performance.

## 1. Introduction

Image denoising is a commonly studied problem in computer vision and has been shown to be important in medical images [1,2], remote sensing images [3], mobile phone images [4], etc. It aims to restore a corrupted image *x* to the ground-truth clean image *y*, which can be modeled as y=x−v, where *v* is the noise. Synthetic noisy images and real-world noisy images are studied in this paper.

Recently, convolutional neural networks (CNNs) have been popularly adapted to image denoising. Zhang et al. [5] proposed a feed-forward denoising a convolutional neural network (DnCNN) with residual learning and batch normalization to remove the additive white Gaussian noise (AWGN). Residual learning here is training networks to estimate the noise of the noisy image and then subtract it to obtain the corresponding clean image. Based on residual learning, CNNs obtained remarkable denoising results, completely exceeding the traditional methods such BM3D [6] and WNNM [7]. Currently, most work focused on designing more effective network modules to enhance denoising performance. For instance, depth networks [8,9,10], width networks [4,11,12,13,14] and attention mechanisms [15,16,17,18,19] were deeply studied. Additionally, some methods employed variations of convolution, such as deformed convolution [20], to improve image denoising.

Despite achieving high performance in image denoising, the aforementioned methods have not fully addressed the limitations of residual learning. Firstly, residual learning ignores the image information, because its optimization target can be expressed as the distance between the estimated noise and the ground-truth noise, formulated as:(1)$$L\left(\theta \right)=L_{f}\left(x-f\left(x,\theta \right),y\right)$$
where $L_{f}$ is an arbitrary loss function, θ is trainable parameters, and f(·) is a neural network. However, generating high-quality denoised images also depends on the image information. Secondly, residual learning does not fully exploit the image self-similarity, which is crucial in image restoration. Existing methods addressed this issue by designing specific function modules, such as non-local mechanisms [17,21,22], which restore one pixel by using its neighboring pixels. However, non-local mechanisms require high computation, especially when considering many neighboring pixels.

This paper aims to solve the aforementioned problems. For the first one, we propose an image-to-image training method that minimizes the Structure Similarity Index Measure (SSIM) loss between denoised images and ground-truth clean images to enable a network to extract the image information effectively. For the second one, we propose using image self-similarity from a training perspective. Specifically, for an image with additive white Gaussian noise (AWGN), the noise distributions in different patches of the image are similar. Thus, we split the noisy image into patches and train the proposed noise estimation path (NEP) to output similar noise estimations among these patches.

Based on the two points, we propose a compositional denoising network (CDN) with an image information path (IIP), a noise estimation path (NEP), and an integration denoising module (IDM). The IIP is optimized using the image-to-image training method to extract image information, while the NEP is trained using image self-similarity to estimate the noise distributions. IDM receives the outputs of IIP and NEP, generates the final estimated noise, and outputs a denoised image via residual subtraction. The main contributions of this paper are as follows:

(1) We suggest two limitations of the commonly used residual learning in image denoising, which are largely ignored in previous research. Our work may inspire further exploration of training methods in image denoising.

(2) We propose to leverage image information and image self-similarity to address the limitations of residual learning. The proposed IIP is optimized with image-to-image training to extract image information, while NEP utilizes similarity-based training to estimate image noise. Our ablation experiments demonstrate the effectiveness of these methods.

(3) Our proposed CDN, built upon IIP and NEP, achieves superior results in image denoising on both synthetic and real-world datasets.

## 2. Related Work

### 2.1. Residual Learning for Image Denoising

Residual learning was proposed in ResNet [23] to solve the performance degradation problem with the increasing network depth. With such a learning strategy, the residual network learns a residual mapping for a few stacked layers. Before ResNet, learning the residual mapping had already been adopted in some low-level vision tasks [24,25]. Zhang et al. [5] extended this concept to image denoising, using a single residual unit to predict the residual image instead of many stacked units. Nowadays, residual learning is widely used in most deep denoising networks. However, residual learning alone may not be sufficient to obtain satisfactory denoising results since image information acquisition is also important. Furthermore, existing residual learning methods do not consider image self-similarity.

### 2.2. Deep Networks for Image Denoising

Over the years, many methods have been proposed for image denoising, including both traditional and deep-learning-based approaches. In this paper, we focus on the deep-learning-based methods. Zhang et al. [5] proposed a deep convolutional neural network (CNN) that goes beyond traditional Gaussian denoisers by utilizing residual learning to learn a residual mapping between noisy and clean images. Ren et al. [9] further extended the use of residual blocks in image denoising through their DN-ResNet, which incorporates dense connections and residual learning. Additionally, Zhang et al. [10] introduced an effective residual block that improves image denoising performance. Tian et al. [12], on the other hand, introduced batch renormalization to deep CNNs for image denoising, which effectively reduces the impact of different batch sizes on training. This study shows the effectiveness of renormalization compared to previous normalization techniques. These approaches demonstrate the efficacy of increasing network depth in addressing image denoising.

However, the increased depth makes models suffer from gradient vanishing or exploding. Hierarchical networks were proposed to use wide network structures to alleviate the problem. Tian et al. [12] utilized a two-path network to increase the width of the network and thus obtained more features. They also proposed a dual denoising network (DudeNet) with two paths and further designed their different functions [11]. Specifically, the top sub-path of DudeNet uses a sparse mechanism to extract global and local features. A non-local hierarchical network (NHNet) [17] used two sub-paths to process different resolutions of the noisy image. For the high-resolution path, it employed a novel upsampling method with a non-local mechanism to obtain effective features. Some U-Net-based networks adopt a three-path structure to improve denoising performance. DHDN [4] replaced the convolution block in the original U-Net [26] with dense blocks and obtained better denoising results. MCU-Net [14] added an extra branch of atrous spatial pyramid pooling (ASPP) based on residual dense blocks. Sub-paths of these models extract different resolution image features, which are fused at the end of the network for denoising. From a frequency domain perspective, some methods [13,27] have employed the multi-level wavelet transform in image denoising and achieved high performance.

This paper utilizes the hierarchical structure to design a network, and the sub-path functions are clearly defined. Specifically, IIP extracts the image information, and NEP estimates the noise distribution.

## 3. The Proposed Method

### 3.1. Network Architecture

The proposed CDN is shown in Figure 1; it consists of three main modules, IIP, NEP, and IDM. Here, C denotes convolution layer, BN denotes batch normalization [28], PR denotes parametric rectified linear unit [29], and R denotes rectified linear unit [30]. Convolution layers in CDN are set kernel size (3×3), stride 1, and padding 1. During training, an input noisy image *x* is divided into four equal patches, $x_{1},x_{2},x_{3},$ and $x_{4}$. This splitting operation aims to utilize the similarity of patches to train NEP. Here, we empirically divide the image into four patches for the following reasons: (1) fewer patches do not take advantage of similarity; and (2) more patches cause fewer noise samples per patch; thus, they can not sufficiently estimate the noise distribution. The training method of IIP is image-to-image. Without loss of generality, we choose the first patch $x_{1}$ and use $y_{1}$ to denote the ground-truth clean image of $x_{1}$. Finally, the denoised $x_{1}$ is output by CDN, noted as $\tilde{x_{1}}$.

When testing, the input of CDN is a complete noisy image, and the output is the denoised image.

#### 3.1.1. Image Information Path (IIP)

IIP consists of one convolution layer and seven DBlocks, as shown in Figure 1. It is proposed to extract the image information. During training, the input of IIP is $x_{1}$. IIP extracts the image features of $x_{1}$, and then the denoised image $x_{1_{c}}$ and noise estimation $x_{1_{n}}$ are obtained based on image features. It is expressed as:(2)$$\begin{array}{cc} x_{1_{c}} & =Conv(IIP\left(x_{1}\right))\\ x_{1_{n}} & =Conv\left(x_{1}\right)-IIP\left(x_{1}\right) \end{array}$$
where *Conv* is the convolution layer changing the number of feature channels. $x_{1_{c}}$ is used to constrain IIP to extract the image information via the image-to-image training method. Therefore, $y_{1}$ is the optimization target of $x_{1_{c}}$, where SSIM is chosen as the loss function. For $x_{1_{n}}$, it is further processed in IDM. The SSIM loss is as follows:(3)$$L_{SSIM}=\frac{2\mu_{x_{1_{c}}}\mu_{y_{1}}+C_{1}}{\mu_{x_{1_{c}}}^{2}+\mu_{y_{1}}^{2}+C_{1}}\times \frac{2\sigma_{x_{1_{c}}y_{1}}+C_{2}}{\sigma_{x_{1_{c}}}^{2}+\sigma_{y_{1}}^{2}+C_{2}}$$
where μ, σ, and $\sigma_{x_{1_{c}}y_{1}}$ denote the mean, standard deviation, and covariance, respectively. C1 and C2 are the image-dependent constants, which provide stabilization against small denominators.

For testing, IIP receives a complete image and outputs its noise based on the image information.

#### 3.1.2. Noise Estimation Path (NEP)

The architecture of NEP is similar to that of IIP. During training, $x_{1},x_{2},x_{3},$ and $x_{4}$ are input into NEP, respectively, and their estimated noise, $n_{1},n_{2},n_{3},$ and $n_{4}$, are outputted. We suggest that $n_{1},n_{2},n_{3},$ and $n_{4}$ should have a similar distribution when noise in the input image is specific. Therefore, Kullback–Leibler divergence (KLD) is used to evaluate the distance of these noise distributions:(4)$$D_{KL}(P∥Q)=\sum P\left(x\right)\log\frac{P(x)}{Q(x)}$$
where *P* and *Q* are probability distributions. The sum of these distances forms the loss function:(5)$$L_{KLD}=\sum_{i=1}^{4}\sum_{j=1,j\neq i}^{4}D_{KL}\left(n_{i}∥n_{j}\right)$$

By minimizing $L_{KLD}$, this similarity-based training method solves the limitation of residual learning.

For testing, NEP receives a complete noisy image and outputs its noise distribution.

#### 3.1.3. Integration Denoising Module (IDM)

IDM is proposed to integrate the outputs from IIP and NEP. It is the U-Net-based network shown in Figure 2. DBlock in IDM is the basic feature extraction block, and PixelShuffle is the upsampling method based on the efficient sub-pixel convolution [31]. IDM outputs the final estimated noise and then obtains the denoised image $\tilde{x_{1}}$ via residual subtraction. L1 loss is used as the loss function of $\tilde{x_{1}}$ and ground-truth clean image $y_{1}$:(6)$$L1=\frac{1}{N}\left|\tilde{x_{1}}-y_{1}\right|$$

### 3.2. Training Loss

The loss function used in this paper consists of three equally important components. Firstly, the image information loss $L_{SSIM}$ is utilized to train IIP to extract image information. Secondly, the noise estimation loss $L_{KLD}$ is employed to train NEP to estimate image noise accurately. Finally, the overall residual loss L1 is used to ensure that the whole network outputs the corresponding clean image of an input noisy image. By combining these three components, CDN can be effectively trained to remove image noise. The training loss can be formulated as follows:(7)$$L=L_{SSIM}+L_{KLD}+L1$$

## 4. Experiments

### 4.1. Datasets

#### 4.1.1. Synthetic Noise Datasets

DIV2K [32] is commonly used in image processing; it contains 800 images for training, 100 for validation, and 100 for testing. We used the training set of DIV2K to train CDN. For testing, we used gray-scale image datasets Set12 [5] and BSD68 [33] and color-scale image datasets Set5 [34] and Kodak24 [35]. Figure 3 shows images of Set12, which contains C.man, House, Peppers, etc. Images in these datasets are all clean and their corresponding synthetic noisy images are generated by adding AWGN. We refer to the AWGN generation algorithm from [5], in which the noise level is determined by the standard deviation σ. Three noise levels, σ=15, σ=25, and σ=50, were chosen to train and test the CDN.

#### 4.1.2. Real-World Noise Datasets

Real-world noise images are directly obtained in the natural environment. Here, we used the training set of the Smartphone Image Denoising DATA (SIDD) sRGB track [36] to train CDN. It contains 160 scene instances captured by five smartphone cameras under different lighting conditions and camera settings. There are two pairs of high-resolution images for each scene instance, and each pair contains one noisy image and its corresponding clean image. In total, 320 pairs of images were used for training. For testing, we used the the SIDD validation set and the Darmstadt Noise Data set (DND) [37]. DND does not provide any training data. It has 50 pairs of images captured by four different consumer cameras for testing. We obtained the PSNR and SSIM results by submitting the denoising images to the official DND website.

### 4.2. Training Setting

CDN is implemented by Pytorch 1.5.1 based on Python 3.5 and Cuda 9.2. Experiments were run on NVIDIA Tesla P100 GPUs. We used the Adam [38] algorithm with an initial learning rate of 0.0002 and a weight decay of 0.0001 to minimize the loss function. The learning rate will decrease with the increment in training epochs. During training, the mini-batch size was set to 64. Data augmentations were adopted to network training, which randomly splits the images into 128 × 128 patches and flips them horizontally and vertically.

### 4.3. Experimental Results

We evaluated the denoising performance of CDN on synthetic and real-world datasets and compared it with some popular methods.

#### 4.3.1. Evaluation Metrics

Peak signal-to-noise ratio (PSNR) was used as the evaluation metric; it is one of the most common indicators for image processing methods. It measures the level of distortion or error between the original image and the reconstructed image by comparing their pixel values. PSNR is calculated based on the mean squared error (MSE) between the two images. The formula for calculating MSE is as follows:(8)$$MSE=\frac{1}{N}\sum {\left(I-R\right)}^{2}$$
where *N* is the total number of pixels in the image and *I* and *R* represent the pixel value of the original image and the reconstructed image, respectively. *PSNR* is then computed as the ratio of the maximum possible pixel value (usually 255 for 8-bit images) to the square root of the *MSE*:(9)$$PSNR=20\cdot \log_{10}\left(MAX\right)-10\cdot \log_{10}\left(MSE\right)$$
where MAX is the maximum possible pixel value of the image. A higher *PSNR* value indicates a lower level of distortion and better image quality.

SSIM is also a widely used method to measure similarity between two images. It is designed to evaluate the perceived quality of images by taking into account their structural information. SSIM compares local patterns of pixel intensities in the reference and distorted images and computes a similarity score ranging from 0 to 1. The formulation of SSIM is described in Section 3.1.1. Similar to PSNR, a higher SSIM value indicates a lower level of distortion and higher image quality.

Consistent with most previous studies, we present the PSNR results for synthetic image denoising and both PSNR and SSIM results for real-world image denoising.

#### 4.3.2. AWGN Denoising

**Gray-scale image**: We first report the training PSNR curve in Figure 4, demonstrating that CDN was well trained and obtains good denoising results on new data from Set12. Table 1 and Table 2 show the synthetic gray-scale noisy image denoising results of different methods on Set12 and BSD68, respectively. CDN outperforms other methods at all noise levels on Set12 and σ=15 and σ=25 on BSD68. Additionally, CDN is a hierarchical network, and compared with other hierarchical networks—U-Net [26], DIDN [39], BRDNet [12] and NHNet [17]—CDN has superior performance.

CDN exhibits the most substantial improvement in denoising the Barbara image in Set12, with improvements of 0.34 at σ=15, 0.53 at σ=25, and 0.9 at σ=50 over the second-best method. As illustrated in Figure 3, Barbara is a highly textured image, which indicates that CDN performs well in preserving image details. Moreover, the superior PSNR results of CDN on other texture-rich images, such as Monarch, Man, and Couple, further support this point. As shown in Figure 5, CDN exhibits the best visual effect in denoising the image Monarch. Overall, these results suggest that CDN is a powerful and effective method for image denoising with the ability to preserve image details and texture.

**Color-scale images**: We also tested CDN on color-scale noisy images. As shown in Table 3, CDN achieves the highest PSNR results on the Kodak24 dataset. The visual comparisons of CDN with other methods on Set5 and Kodak24 are shown in Figure 6 and Figure 7, respectively. The results indicate that CDN outperforms the other methods and can recover cleaner images.

#### 4.3.3. Real-World Image Denoising

While the AWGN task can provide some insight into the effectiveness of a denoising method, its limitation is clear. Real-world noise is more complicated and unpredictable, so evaluating denoising methods on real-world noisy images is more meaningful. To assess CDN’s performance in real-world image denoising, we used the SIDD validation set and DND. Table 4 lists denoising results of different methods, where CDN achieves the best PSNR and SSIM results on the SIDD validation set and competitive performance on DND. Figure 8 shows some denoised images of CDN on the SIDD dataset, indicating that CDN successfully removes noise. These results demonstrate that CDN is also useful in denoising real-world images and has practical application value.

### 4.4. Ablation Experiments

The effectiveness of CDN in denoising relies on two key components: IIP, which extracts image information through image-to-image training, and NEP, which estimates noise through image self-similarity training. In this section, we study the contributions of these components in detail and demonstrate their effectiveness in improving denoising results.

#### 4.4.1. Role of IIP

IIP in CDN is crucial for extracting image information, and it is necessary to determine whether this information improves the denoising performance. We first conducted an ablation experiment by removing it from a pretrained CDN model, named CDN-IIP. This was achieved by replacing the output of IIP with a zero matrix of the same size. Figure 9 shows the denoising results of CDN-IIP. Compared to CDN, the denoised images produced by CDN-IIP are significantly blurred and lack details, demonstrating the importance of image information in restoring image details. Then, we further evaluated IIP by removing IIP and retraining CDN, denoted CDN-IIP(R) in Table 5. The results show that removing IIP leads to a decrease in both PSNR and SSIM, further confirming the effectiveness of IIP.

IIP uses SSIM loss as the loss function because it provides a comprehensive measure of image similarity based on brightness, contrast, and structure. We compared the denoising results using different loss functions, including L1 loss and mean square error (MSE) loss, and found that the SSIM loss provides better denoising performance, as shown in Table 6.

#### 4.4.2. Role of NEP

NEP provides noise estimation of an noisy image. CDN-NEP in Figure 9 denotes CDN cutting off NEP. It can be seen that although the denoised image of CDN-NEP contains sufficient image information, the noise is obviously not removed well. Therefore, the noise distribution estimated from NEP is essential for removing noise. Similar to the study of IIP, we also report the PSNR and SSIM results of retained CDN-NEP. The results of CDN-NEP(R) on Set12 are listed in Table 5, and they are are significantly lower than that of CDN. This demonstrates that using the estimated noise distribution can improve denoising performance.

#### 4.4.3. Role of Training Methods

CDN solves the limitations of traditional residual learning by using the image-to-image training method to train IIP and the similarity-based training method to train NEP. Here, we study the effect of these training methods on the denoising performance. CDN-SSIM in Table 5 denotes that CDN is trained without the image-to-image training method, which is implemented by removing the SSIM loss during training. Similarly, CDN-KLD denotes that CDN is trained without the similarity-based training method. CDN-SSIM-KLD denotes CDN is trained without either training method, which can be considered ordinary residual learning. The results in Table 5 show CDN that performs comprehensively better than the other methods. In particular, CDN significantly outperforms CDN-SSIM-KLD, indicating that considering image information and self-similarity improves residual learning.

The input image is divided into patches during training. The number of patches also affects the denoising performance. Table 7 lists the PSNR denoising results of different numbers of patches, which shows that too many patches leads to noise performance degradation. The reason is that many patches cause small patch sizes and thus an individual patch can not contain enough image information. In order to concisely describe the proposed model, four patches were selected in this paper.

## 5. Discussion

Deep-learning-based image denoising methods are increasingly popular among researchers due to their ease of implementation and fast processing speed. While most research focuses on improving network architecture, potential limitations in the commonly used residual learning method are often neglected. This paper points out two limitations of the residual learning and proposes a novel denoising CDN to solve them. We conducted comparison experiments between the proposed methods (CDN) and original residual learning (CDN-SSIM-KLD), which demonstrated that our solution significantly improves denoising performance.

IIP and NEP in CDN with their training methods aim to solve the limitations of residual learning that do not consider image information and image self-similarity, respectively. Specifically, IIP is trained to extract image information using an image-to-image approach, while NEP estimates image noise by leveraging image self-similarity. To explicate their functions, we conducted corresponding experiments. Firstly, we observed that removing either IIP or NEP could result in an increase in PSNR and SSIM, which demonstrated their significance in image denoising. Secondly, we visualized the denoised images of CDN-IIP and CDN-NEP. Results revealed that CDN without IIP was able to successfully remove noise but failed to preserve fine image details. The inverse situation can be found in CDN without NEP. These results illustrate that IIP and NEP achieve the expected function and improve the residual learning.

However, there are still some limitations in our work. The same architecture is used for both IIP and NEP, and exploring different architectures could potentially improve the performance of the network. For example, vision-transformer-based image denoising networks have achieved state-of-the-art performance [45,46]. Incorporating a vision transformer as the backbone of IIP or NEP may enhance the denoising performance of CDN. In addition, although SSIM loss shows better results than L1 and MSE for the image information loss, there might be other loss functions that could be more effective. These concerns will be explored in future studies.

## 6. Conclusions

This paper introduces a novel denoising network, CDN, which aims to overcome the two limitations of residual learning in image denoising. Firstly, residual learning fails to fully consider image information. This is tackled by training IIP in CDN to extract image information through the proposed image-to-image method. Secondly, residual learning does not take into account image self-similarity. To solve this issue, we propose a similarity-based training method to train NEP in CDN to estimate image noise. Consequently, CDN can successfully remove image noise using the extracted image information from IIP and noise estimation from NEP. Experimental results show that CDN achieves superior performance in both synthetic and real-world image denoising. Besides the high performance, we also discuss the potential limitations in our work. While previous studies primarily focused on improving network architecture, we present a novel perspective that improves the training method for enhanced denoising results. It may encourage further exploration into the effectiveness of training methods in image denoising research. 

## Figures and Tables

**Figure 1 sensors-23-05915-f001:**
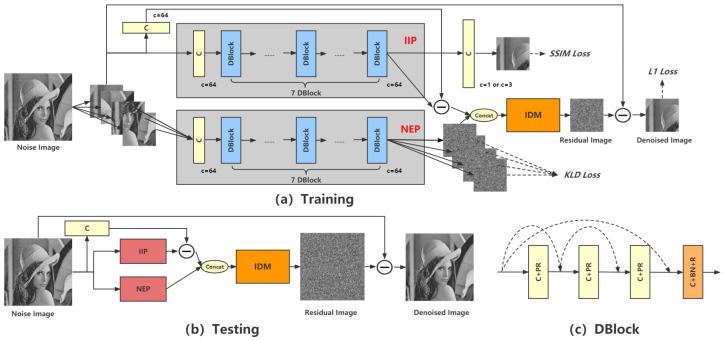
The architecture of CDN. It contains three main modules, IIP, NEP, and IDM. (**a**) The training stage. A noisy image is split into several patches. The training target is removing the first patch’s noise. Other patches are used in self-similarity training for NEP. IIP is optimized to extract image information in the image-to-image training method. NEP is trained by considering image self-similarity, which estimates the noise distribution. IDM integrates the output of IIP and NEP to obtain the final estimated noise. It is shown in Figure 2. (**b**) The testing stage. It removes the noise of a complete image. (**c**) The basic CNN block of CDN, DBlock.

**Figure 2 sensors-23-05915-f002:**
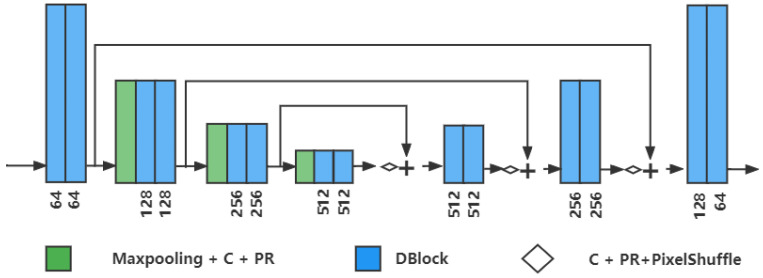
The integration denoising module (IDM). It uses UNet-based architecture, and the number of output channels of each DBlock is shown at the bottom. DBlock is shown in Figure 1c.

**Figure 3 sensors-23-05915-f003:**
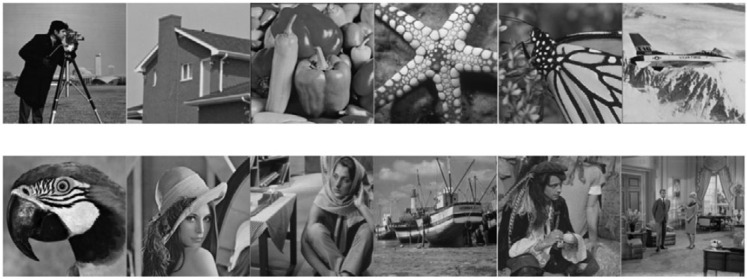
Images in Set12, which are C.man, House, Peppers, Starfish, Monarch, Airplane, Parrot, Lena, Barbara, Boat, Man, and Couple, in order.

**Figure 4 sensors-23-05915-f004:**
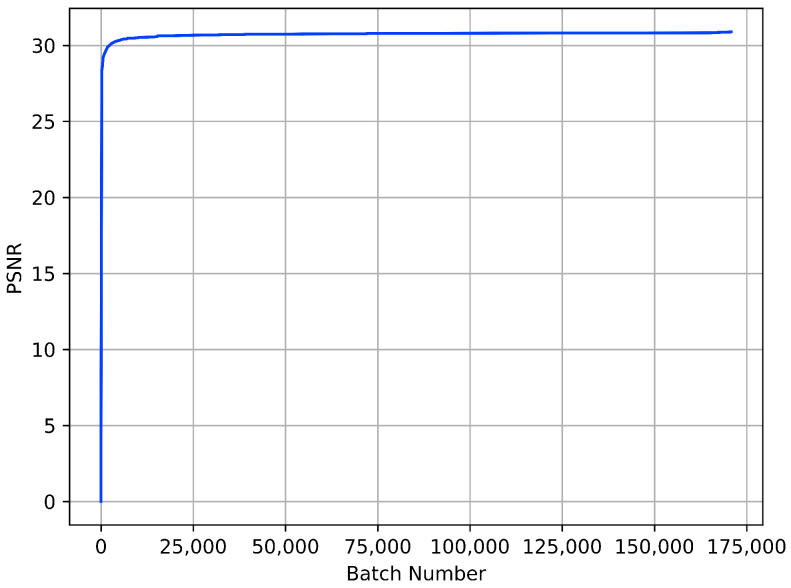
Training PSNR curve for CDN in AWGN denoising. The training dataset is the DIV2K training set and PSNR results are computed on Set12 at noise level σ=25.

**Figure 5 sensors-23-05915-f005:**
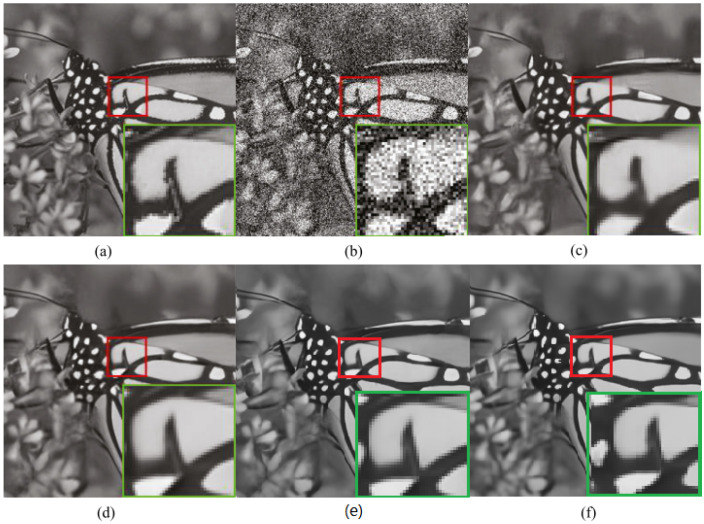
PSNR results of the image Monarch from Set12 with noise level σ=50. (**a**) Clean image, (**b**) noisy image/14.71 dB, (**c**) DnCNN [5]/26.78 dB, (**d**) BRDNet [12]/26.97 dB, (**e**) MHCNN [42]/27.12 dB, and (**f**) CDN/27.21 dB.

**Figure 6 sensors-23-05915-f006:**
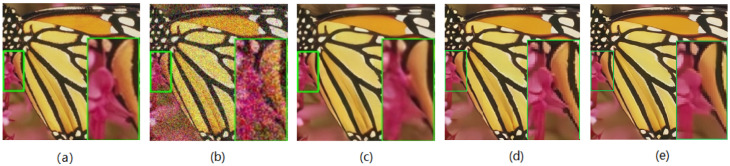
Denoising result of Butterfly from Set5 at noise level σ=50. (**a**) Clean image, (**b**) noisy image, (**c**) VDN, (**d**) NHNet, (**e**) CDN.

**Figure 7 sensors-23-05915-f007:**
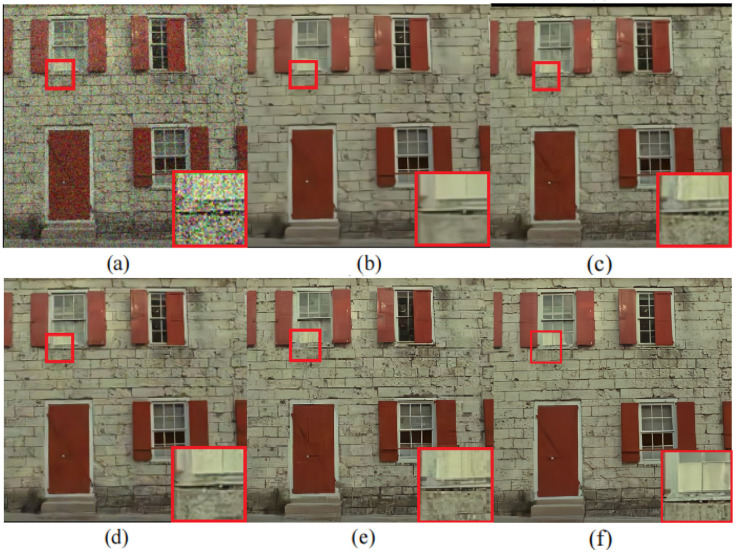
Denoising result on the department wall from Kodak24 at noise level σ=50. (**a**) Noisy image, (**b**) DnCNN/25.80 dB, (**c**) BRDNet/26.33 dB, (**d**) FFDNet/26.13 dB, (**e**) NHNet/26.49 dB, and (**f**) CDN/29.53 dB.

**Figure 8 sensors-23-05915-f008:**
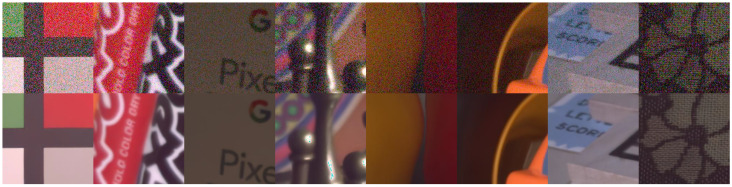
Denoising results of CDN on SIDD.

**Figure 9 sensors-23-05915-f009:**
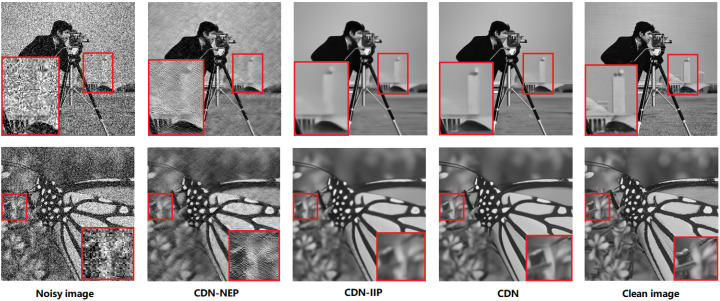
Visual denoising results comparison of the ablation models of CDN. Images are from Set12. The left and right columns show the images with noise level σ=50 and the clean images, respectively. The middle three columns list the denoised images from CDN and its ablation variants.

**Table 1 sensors-23-05915-t001:** PSNR (dB) results of different networks on Set12 at noise levels of 15, 25, and 50.

Image	*C.man*	*House*	*Peppers*	*Starfish*	*Monarch*	*Airplane*	*Parrot*	*Lena*	*Barbara*	*Boat*	*Man*	*Couple*	*Average*
**Noise level**	** σ=15 **												
**BM3D [6]**	**31.91**	**34.93**	**32.69**	**31.14**	**31.85**	**31.07**	**31.37**	**34.26**	**33.10**	**32.13**	**31.92**	**32.10**	**32.37**
**DnCNN [5]**	**32.61**	**34.97**	**33.30**	**32.20**	**33.09**	**31.70**	**31.83**	**34.62**	**32.64**	**32.42**	**32.46**	**32.47**	**32.86**
**FFDNet [40]**	**32.43**	**35.07**	**33.25**	**31.99**	**32.66**	**31.57**	**31.81**	**34.62**	**32.54**	**32.38**	**32.41**	**32.46**	**32.77**
**ResDNN [8]**	**32.73**	**34.99**	**33.23**	**32.11**	**33.20**	**31.65**	**31.87**	**34.57**	**32.56**	**32.39**	**32.42**	**32.43**	**32.85**
**U-Net [41]**	**32.33**	**34.79**	**33.16**	**32.00**	**32.94**	**31.64**	**31.84**	**34.46**	**32.43**	**32.30**	**32.34**	**32.31**	**32.71**
**ADNet [15]**	** 32.81 **	**35.22**	** 33.49 **	**32.17**	**33.17**	**31.86**	**31.96**	**34.71**	**32.80**	**32.57**	**32.47**	**32.58**	**32.98**
**DudeNet [11]**	**32.71**	**35.13**	**33.38**	**32.29**	**33.28**	**31.78**	**31.93**	**34.66**	**32.73**	**32.46**	**32.46**	**32.49**	**32.94**
**BRDNet [12]**	**32.80**	**35.27**	**33.47**	**32.24**	**33.35**	**31.82**	**32.00**	**34.75**	**32.93**	**32.55**	**32.50**	**32.62**	**33.03**
* **NHNet** * ** [17]**	** 32.95 **	** 35.40 **	** 33.60 **	** 32.36 **	** 33.55 **	** 31.98 **	** 32.10 **	** 34.80 **	** 33.14 **	** 32.65 **	** 32.57 **	** 32.69 **	** 33.15 **
* **CDN** *	**32.73**	** 35.70 **	**33.42**	** 32.40 **	** 33.57 **	** 31.87 **	** 32.02 **	** 34.91 **	** 33.48 **	** 32.73 **	** 32.58 **	** 32.73 **	** 33.18 **
**Noise level**	** σ=25 **												
**BM3D** [6]	**29.45**	**32.85**	**30.16**	**28.56**	**29.25**	**28.42**	**28.93**	**32.07**	** 30.71 **	**29.90**	**29.61**	**29.71**	**29.97**
**DnCNN** [5]	**30.18**	**33.06**	**30.87**	**29.41**	**30.28**	**29.13**	**29.43**	**32.44**	**30.00**	**30.21**	**30.10**	**30.12**	**30.43**
**FFDNet** [40]	**30.10**	**33.28**	**30.93**	**29.32**	**30.08**	**29.04**	**29.44**	**32.57**	**30.01**	**30.25**	**30.11**	**30.20**	**30.44**
**ResDNN** [8]	**30.17**	**32.99**	**30.73**	**29.24**	**30.30**	**29.00**	**29.38**	**32.31**	**29.70**	**30.11**	**30.04**	**29.96**	**30.33**
**U-Net** [41]	**30.18**	**33.18**	**30.91**	**29.38**	**30.41**	**29.18**	**29.57**	**32.59**	**30.19**	**30.25**	**30.10**	**30.14**	**30.51**
**ADNet** [15]	**30.34**	**33.41**	**31.14**	**29.41**	**30.39**	**29.17**	**29.49**	**32.61**	**30.25**	**30.37**	**30.08**	**30.24**	**30.58**
**DudeNet** [11]	**30.23**	**33.24**	**30.98**	**29.53**	**30.44**	**29.14**	**29.48**	**32.52**	**30.15**	**30.24**	**30.08**	**30.15**	**30.52**
**BRDNet** [12]	** 31.39 **	**33.41**	**31.04**	**29.46**	**30.50**	**29.20**	**29.55**	**32.65**	**30.34**	**30.33**	**30.14**	**30.28**	**30.61**
* **NHNet** * ** [17]**	** 30.49 **	** 33.65 **	** 31.20 **	** 29.72 **	** 30.68 **	** 29.34 **	** 29.65 **	** 32.76 **	**30.70**	** 30.44 **	** 30.20 **	** 30.40 **	** 30.77 **
* **CDN** *	**30.45**	** 33.86 **	**31.12**	** 29.78 **	** 30.91 **	** 29.29 **	** 29.67 **	** 32.95 **	** 31.24 **	** 30.60 **	** 30.27 **	** 30.52 **	** 30.89 **
**Noise level**	** σ=50 **												
**BM3D [6]**	**26.13**	**29.69**	**26.68**	**25.04**	**25.82**	**25.10**	**25.90**	**29.05**	** 27.22 **	**26.78**	**26.81**	**26.46**	**26.72**
**DnCNN [5]**	**27.03**	**30.00**	**27.32**	**25.70**	**26.78**	**25.87**	**26.48**	**29.39**	**26.22**	**27.20**	**27.24**	**26.90**	**27.18**
**FFDNet [40]**	**27.05**	**30.37**	**27.54**	**25.75**	**26.81**	**25.89**	**26.57**	**29.66**	**26.45**	**27.33**	**27.29**	**27.08**	**27.32**
**ResDNN [8]**	**26.63**	**29.27**	**26.68**	**25.31**	**26.27**	**25.35**	**26.01**	**28.80**	**24.48**	**26.72**	**26.90**	**26.25**	**26.56**
**U-Net [41]**	**27.42**	**30.48**	**27.67**	**25.92**	**26.94**	**25.89**	**26.66**	** 29.84 **	**27.02**	**27.42**	**27.30**	**27.17**	**27.48**
**ADNet [15]**	**27.31**	**30.59**	**27.69**	**25.70**	**26.90**	**25.88**	**26.56**	**29.59**	**26.64**	**27.35**	**27.17**	**27.07**	**27.37**
**DudeNet [11]**	**27.22**	**30.27**	**27.51**	**25.88**	**26.93**	**25.88**	**26.50**	**29.45**	**26.49**	**27.26**	**27.19**	**26.97**	**27.30**
**BRDNet [12]**	**27.44**	**30.53**	**27.67**	**25.77**	**26.97**	**25.93**	**26.66**	**29.73**	**26.85**	**27.38**	**27.27**	**27.17**	**27.45**
* **NHNet** * ** [17]**	** 27.54 **	** 30.85 **	** 27.84 **	** 26.24 **	** 27.10 **	** 26.00 **	** 26.76 **	**29.83**	**27.19**	** 27.46 **	** 27.32 **	** 27.28 **	** 27.62 **
* **CDN** *	** 27.70 **	** 31.26 **	** 27.82 **	** 26.29 **	** 27.23 **	** 26.06 **	** 26.88 **	** 30.07 **	** 28.12 **	** 27.65 **	** 27.42 **	** 27.52 **	** 27.83 **

**Table 2 sensors-23-05915-t002:** Results of different networks on BSD68.

Network	*BM3D* [6]	*DnCNN* [5]	*FFDNet* [40]	*ADNet* [15]	*DudeNet* [11]	*BRDNet* [12]	*U-Net* [41]	*RIDNet* [19]	*NHNet*	*CDN*
ine **σ=15**	**31.07**	**31.72**	**31.62**	**31.74**	**31.78**	**31.79**	**31.54**	**31.81**	** 31.85 **	** 31.89 **
ine **σ=25**	**28.57**	**29.23**	**29.19**	**29.25**	**29.29**	**29.29**	**29.13**	**29.34**	** 29.37 **	** 29.44 **
ine **σ=50**	**25.62**	**26.23**	**26.30**	**26.29**	**26.31**	**26.36**	**26.39**	** 26.40 **	** 26.43 **	**26.39**

**Table 3 sensors-23-05915-t003:** Color image denoising results of different networks.

Dataset	*Method*	σ=15	σ=25	σ=50
Set5	* **CBM3D** * ** [6]**	**33.42**	**30.92**	**28.16**
	* **FFDNet** * ** [40]**	**34.30**	**32.10**	**29.25**
	* **VDN** * ** [43]**	**34.34**	**32.24**	**29.47**
	* **NHNet** * ** [17]**	** 34.80 **	** 32.56 **	** 29.64 **
	* **CDN** *	** 34.70 **	** 32.58 **	** 29.66 **
Kodak24	* **CBM3D** * ** [6]**	**34.28**	**31.68**	**28.46**
	* **FFDNet** * ** [40]**	**34.55**	**32.11**	**28.99**
	* **DnCNN ** *	**34.73**	**32.23**	**29.02**
	* **ADNet** * ** [15]**	**34.76**	**32.26**	**29.10**
	* **DudeNet** * ** [11]**	**34.81**	**32.26**	**29.10**
	* **BRDNet** * ** [12]**	**34.88**	**32.41**	**29.22**
	* **NHNet** * ** [17]**	** 35.02 **	** 32.54 **	** 29.41 **
	* **CDN** *	** 35.05 **	** 32.57 **	** 29.54 **

**Table 4 sensors-23-05915-t004:** Denoising results of different networks on real-world noise datasets.

Test Data	SIDD Validation							
Method	BM3D [6]	WNNM [7]	CBDNet [44]	RIDNet [19]	VDN [43]	MHCNN [42]	CDN	
PSNR	25.65	25.78	38.68	38.71	39.28	** 39.06 **	** 39.36 **	
SSIM	0.685	0.685	0.809	** 0.914 **	0.909	** 0.914 **	** 0.918 **	
**Test Data**	**DND**							
Method	BM3D [6]	WNNM [7]	CBDNet [44]	RIDNet [19]	VDN [43]	PAN-Net [18]	MHCNN [42]	CDN
PSNR	34.51	34.67	38.06	39.26	39.38	** 39.44 **	** 39.52 **	** 39.44 **
SSIM	0.851	0.865	0.942	** 0.953 **	** 0.952 **	** 0.952 **	0.951	0.951

**Table 5 sensors-23-05915-t005:** Ablation experiment results on Set12.

Method	σ=15		σ=25		σ=50	
	* **PSNR** *	* **SSIM** *	* **PSNR** *	* **SSIM** *	* **PSNR** *	* **SSIM** *
*CDN-IIP(R)*	33.14	0.9080	30.82	0.8712	27.77	0.8050
*CDN-NEP(R)*	33.02	0.9058	30.57	0.8659	27.54	0.7991
*CDN-SSIM*	33.10	0.9075	30.80	0.8709	27.76	0.8052
*CDN-KLD*	33.15	0.9083	30.83	0.8709	27.78	0.8051
*CDN-SSIM-KLD*	33.05	0.9069	30.78	0.8703	27.75	0.8049
*CDN*	**33.18**	**0.9089**	**30.89**	**0.8724**	**27.83**	**0.8074**

**Table 6 sensors-23-05915-t006:** Comparison results of SSIM and other loss functions in IIP on Set12 at noise level σ=25.

Loss Function	*L1*	*MSE*	*SSIM*
*PSNR*	30.86	30.85	**30.89**

**Table 7 sensors-23-05915-t007:** PSNR results of CDN on Set12 with different training patches.

Patches	σ=15	σ=25	σ=50
*4*	**33.18**	30.89	27.83
*9*	**33.18**	**30.90**	**27.84**
*16*	33.14	30.82	27.69

## Data Availability

Datasets used in this paper are open access and are available from: DIV2K is openly available in “NTIRE 2017 challenge on single image super-resolution: Dataset and study”, reference number [32]; Set12 is openly available in “Beyond a Gaussian Denoiser: Residual Learning of Deep CNN for Image Denoising”, reference number [5]; BSD68 is openly available in “Fields of Experts: a framework for learning image priors”, reference number [33]; Set5 is openly available in “Accurate Image Super-Resolution Using Very Deep Convolutional Networks”, reference number [34]; Kodak24 is openly available in “Kodak lossless true color image suite: PhotoCD PCD0992” at url: http://r0k.us/graphics/kodak.182 (accessed on 14 June 2023), reference number [35]; SIDD is openly available in “A High-Quality Denoising Dataset for Smartphone Cameras”, reference number [36]; DND is openly available in “Benchmarking Denoising Algorithms with Real Photographs”, reference number [37]. A preprint has previously been published in arXiv by Zhang et al. [47]. Our code will be released on https://github.com/JiaHongZ/CDN (accessed on 14 June 2023).

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
