# Peer review of "Considering Image Information and Self-Similarity: A Compositional Denoising Network"

_sensors, 2023, doi:10.3390/s23135915_

Round 1

Reviewer 1 Report

The paper proposes  a new denoising network, CDN, to solve some limitations of residual learning in image denoising problems. The results are better than existing methods. Tests are made on both grey and color images; there are used several state-of-art methods. The training methods included seems to facilitate the process of removing noise from synthetic and real-world images.

Discussions of included of experiments are included. Advantages and disadvantages of the methods included are presented.

Observation: Please use Caps for  pytorch (PyTorch), and similar.

Author Response

Point 1: Please use Caps for pytorch (PyTorch), and similar.

Response 1: Thank you for this comment. We have updated the manuscript and used Caps for Pytorch, Python, Cuda and so on.

Reviewer 2 Report

The researchers introduced a novel approach called the compositional denoising network (CDN), which is composed of two embedded networks: the image information path (IIP) and the noise estimation path (NET). The IIP is responsible for extracting image information, while the NEP leverages image self-similarity. By combining the IIP and NEP paths within the CDN, the authors achieved superior performance compared to other CNN-based methods in image denoising. They conducted experiments on various image sets to demonstrate the effectiveness of their proposed approach. Furthermore, the authors performed an ablation experiment to highlight the significance of the IIP and NEP in the denoising process. Overall, the paper is well-written, well-organized, and, in my opinion, can be accepted with minor revisions. I have provided my comments and suggestions below.

1. In the description of Line 118, I assumed that the author divided the images into four equal patches. If so, please state that clearly.

2. The author should illustrate the procedure to reform the full image from the patches in the IDM network.

3. Considering the naïve readers, I suggest providing a brief description of the PSNR evaluation metrics.

4. The author only used PSNR as the evaluation metric. Though the proposed method uses SSIM loss for the patches in the IIP network, I would suggest adding the SSIM of full images as another evaluation metric to make the comparison more comprehensive.  

Please review the paper to avoid any grammatical errors and typos.
